# Factors Influencing Fatalities or Severe Injuries to Pedestrians Lying on the Road in Japan: Nationwide Police Database Study

**DOI:** 10.3390/healthcare9111433

**Published:** 2021-10-24

**Authors:** Mirae Koh, Masahito Hitosugi, Eiko Kagesawa, Takahiro Narikawa, Kohei Takashima

**Affiliations:** 1Department of Legal Medicine, Shiga University of Medical Science, Tsukinowa, Seta, Otsu, Shiga 520-2192, Japan; kohmire@yahoo.co.jp (M.K.); takachan@belle.shiga-med.ac.jp (K.T.); 2Institute for Traffic Accident Research and Data Analysis, Sarugaku, Kanda, Chiyoda-ku, Tokyo 101-0064, Japan; e_kagesawa@itarda.or.jp (E.K.); narikawa.t@mazda.co.jp (T.N.)

**Keywords:** pedestrian, fatality, lying on the road, nationwide data, injury, safety

## Abstract

To help reduce the number of pedestrians lying on the road suffering fatal or severe injuries as a result of vehicle collisions, we investigated the influencing factors. We conducted an analysis of the records of the Institute for Traffic Accident Research and Data Analysis Japan between 2012 and 2018; we found that 2452 pedestrians lying on the road were involved in collisions (797 fatalities, 784 severely injured, 871 mildly injured). Multivariate logistic regression analysis identified the following as major factors that positively influenced the fatalities: head or neck injuries (odds ratio [OR], 90.221); trunk injuries (OR, 71.040); trucks as offending vehicle (OR, 2.741); collision velocity of 10–20 km/h (OR, 31.794), 20–30 km/h (OR, 2.982), 30–40 km/h (OR, 8.394), 40–50 km/h (OR, 16.831), and >50 km/h (OR, 18.639); and hit-and-run cases (OR, 1.967). The following had a positive influence on severe injuries: trunk injuries (OR, 4.060); collision velocity of 10–20 km/h (OR, 2.540), 20–30 km/h (OR, 3.700), 30–40 km/h (OR, 5.297), 40–50 km/h (OR, 5.719), and ≥50 km/h (OR, 5.244); and hit-and-run cases (OR, 2.628). Decreasing the collision velocity, avoiding collisions to the head and neck or trunk, and preventing hit-and-run cases would be effective in reducing fatal or severe injuries to pedestrians lying on the road.

## 1. Introduction

Every year, 1.35 million people die on the world’s roads; road crashes kill one person every 24 s [1]. Most of those deaths and injuries are considered preventable. Road safety is a critical issue for both sustainable development and human rights. The United Nations set sustainable development goals for 2030; the goals include everyone having access to safe, affordable, accessible and sustainable transport systems with improving road safety [1]. The General Assembly of the United Nations set a new target for the international community: reducing the number of road deaths by 50% by 2030 as the prime objective of the new decade of action for road safety, 2021–2030 [2].

The overwhelming majority of road deaths involve vulnerable road users; pedestrian fatalities accounted for 23% of all road users’ deaths [3]. Thus, greater efforts by all countries are needed to decrease pedestrian fatalities. To make effective preventive measures, it is necessary to collect, monitor, and analyze data related to pedestrian deaths. The typical situation with vehicle-pedestrian collisions is a pedestrian aiming to cross a road and being hit by a vehicle. 

For such pedestrians, the factors influencing the injury severities or fatalities have been examined using collision databases. For collisions that occurred at intersections in the US state of Illinois, pedestrian age, vehicle type, point of first contact, and weather conditions significantly affected pedestrian injury severity [4]. A collision investigation in Changsha in central China found that the risks of pedestrians sustaining injuries with an Abbreviated Injury Scale score of three or more were age and impact speed [5]. One study conducted in several parts of Indonesia revealed that crash location, type of vehicle, pedestrian age, road hierarchy, and driving license ownership were significant influencing factors for pedestrian injury severity; however, the selected variables differed in each area [6]. One report in Iran found that jaywalking or waiting at the roadside in poorly lit locations substantially increased pedestrian fatality risk [7]. One investigation studied vehicle-pedestrian collisions at mid-blocks in the United States using various variables that affected injury severity [8]. It found that the pedestrians’ age and gender, road speed limit, number of lanes, light, and road surface conditions affected injury severity by influencing the pedestrians’ pre-crash behavior. Those authors underlined the importance of investigating such behavior. One US study in Louisiana examined pedestrians’ behavior and the risks affecting injury levels when pedestrians were under the influence of alcohol or drugs [9]. The elucidated risks were intersection collision at business/industrial locations, mid-block collisions on undivided roadways at residential and business/residential locations, segment related collisions associated with a pedestrian standing in the road, open country collisions with no lighting at night, and pedestrian violation related collisions on divided roadways [9]. 

However, there are also situations where a vehicle hits a pedestrian lying on the road. The reasons for pedestrians being run over in that way have been identified as follows: a previous collision and subsequent falling to the ground; falling to the ground because of acute sickness or intoxication; and deliberately lying on the road to cause self-harm [10]. Such collisions have been particularly examined by forensic practitioners; however, there are no related comprehensive statistics for Japan. Thus, many practitioners have stressed the importance of collecting data about collisions involving pedestrians lying on the road in that country.

The Institute for Traffic Accident Research and Data Analysis, Japan (ITARDA) maintains a large, all-inclusive database of motor vehicle collisions (MVCs) using data provided by Japan’s National Police Agency (NPA). Since 2012, the database has included the category of pedestrians lying on the road to promote in-depth investigation of vehicle-pedestrian collisions. A detailed investigation about such pedestrians has been conducted. According to the database, pedestrians lying on the road accounted for 8.3% of all pedestrian fatalities from 2012 to 2016; thus, the numbers are considerable [11]. Among casualties to pedestrians lying on the road, 33.0% were fatalities, 30.8% severe injuries, and 36.2% mild injuries. It is surprising that although victims had been lying on the road and involved in collisions, two-thirds avoided fatalities and approximately one-third suffered only minor injuries. 

To reduce pedestrian fatalities, it is necessary to make efforts to prevent such fatalities. However, the factors influencing fatalities among all casualties for pedestrians lying on the road have not been determined. Thus, in-depth investigations are required to supply detailed information, such as the following: road environment; victim’s age, stature, and injuries; type of offending vehicle and collision velocity; and part of vehicle causing the injury. Towards reducing fatalities of pedestrians lying on the road, we investigated the related influencing factors.

## 2. Materials and Methods

### 2.1. Study Design

We extracted data from ITARDA records. The ITARDA database includes details of MVCs provided by the NPA occurring on highways or roads open to the public involving one or more vehicles. Using this database, we chose vehicle-pedestrian collisions for 2012–2018. Taking into account the actions of pedestrians immediately before the collision, we selected for analysis cases in which the victims had been lying on the road. 

### 2.2. Collected Data

We examined the following factors in each case:(1)Injury levels

According to the diagnosis of physicians, we categorized the pedestrians’ injury levels as death, severe injury, or mild injury. We defined death as the fatality occurring within 24 h of the collision. A pedestrian suffering from injuries that needed 30 or more days’ treatment was defined as severe injury; we defined under 30 days as mild injury.

(2)Most severely injured body region

According to the medical data, we determined the body region with the most severe injury. We classified the regions as head and neck, trunk, or extremities. If the victim had suffered severe injury in all body regions, we categorized that as all.

(3)Occurrence of collision

We determined the date and time of the collisions. The day was classified as weekday or weekend. We categorized time as day (sunrise to sunset) or night. The place of the collision was classified as urban or rural. The urban area was defined that buildings or houses were shown along with the road for more than 500 m. We categorized the distance from the victim’s home to the place of collision as ≤1 km, 1–2 km, or >2 km. The place of collision was classified as follows: intersection; first lane; second lane or above; non-separated road; or other.

(4)Characteristics of collision

The offending vehicle was categorized as passenger car, cargo, light passenger car, light cargo, truck, motorcycle, or other. We differentiated ordinary and light vehicles according to the engine displacement: the former was >660 cc and the latter ≤660 cc. For passenger cars only, we classified the minimum height of the vehicle floor as <18 cm or ≥18 cm. We categorized the movement of the vehicle as going straight, starting, turning left, turning right, reversing, or other. The vehicle speed immediately before the collision, which had been determined by the police, was classified in 10 km/h units. We categorized the part of the vehicle that struck the pedestrian as wheel or other. We determined whether or not the collision was a hit-and-run case. We defined such a case as follows: the driver of the offending vehicle drove away after striking the pedestrian without offering any aid.

### 2.3. Statistical Analysis

Data were summarized as values with proportions or frequencies for categorical variables. For continuous variables, we used the mean ± standard deviation for values that followed a normal distribution. We applied Chi-square tests to compare the prevalence between two groups. To identify differences in the values between two groups, we used Student’s t test for values that followed a normal distribution. To identify variables that were independently associated with fatality or having severe injuries, we performed multivariate logistic regression analyses. We applied the Hosmer–Lemeshow test to determine goodness-of-fit of the regression models: with that test, higher probability indicates better fit. Additionally, we calculated pseudo R^2^ (Nagelkerke R^2^) as an index of the degree of proportion explainable by the regression equation: a larger R^2^ indicates a better model. We considered a *p* value of 0.05 or less to be statistically significant. The analyses were performed using SPSS version 23 (IBM, Chicago, IL, USA).

This study was performed with the approval of the NPA. A draft of the manuscript was reviewed by the NPA, and we received their permission for submission (2021-09-15).

## 3. Results

### 3.1. General Characteristics

For the 7-year study period, the database registered 389,975 pedestrian casualties and 10,233 fatalities. From that data set, we selected for analysis the 2452 pedestrians who had been lying on the road; there were 797 fatalities and 1655 casualties (784 severe injuries, and 871 mild injuries). Among all pedestrian casualties registered on the database during the study period, pedestrians lying on the road accounted for 7.8% of fatalities and 0.4% of casualties.

The mean age of the victims was 49 years, and males were dominant (Table 1). Over one-third of the collisions occurred at the weekend, and more than 80% took place at night. The collisions mostly occurred in rural areas and within 1 km of the victims’ homes. The most common type of offending vehicle was a passenger car, and the collision mostly occurred when the vehicle was driving straight. 

### 3.2. Comparison of Fatal and Non-Fatal Cases

We divided the collisions into fatal or non-fatal groups. The victim and collision characteristics were compared between those two groups (Table 1). With fatal cases, the victims were significantly older (*p* < 0.001), and they suffered more from head and neck or trunk injuries (*p* < 0.001). The collisions occurred significantly more often at night (*p* < 0.001), in rural areas (*p* = 0.006), within 1 km of the victim’s home (*p* = 0.003), and in the first lane of the road (*p* < 0.001). Regarding the offending vehicles in fatal cases, the prevalence of trucks was significantly greater (*p* < 0.001), and the vehicle was mostly driving straight (*p* < 0.001). In terms of collision velocity with fatal cases, over 30 km/h accounted for 71.3%; ≤30 km/h accounted for 66.5% with non-fatal cases. Hit-and-run cases were significantly more frequent with fatal than with non-fatal cases (*p* < 0.001). 

To identify the variables that were independently associated with fatalities, we undertook multivariate logistic regression analysis using the forced input method. The following had a positive influence on fatalities: age (odds ratio [OR], 1.030); head or neck injuries (OR, 90.221), trunk injuries (OR, 71.040), and all body regions (OR, 21,530.778) with extremities as the reference; weekend (OR, 1.320); night (OR, 2.003); truck as offending vehicle (OR, 2.741); collision velocity of 10–20 km/h (OR, 1.794), 20–30 km/h (OR, 2.982), 30–40 km/h (OR, 8.394), 40–50 km/h (OR, 16.831), and ≥50 km/h (OR, 18.639); hit-and-run (OR, 1.967); and height of the passenger vehicle floor of under 18 cm (OR, 1.984). The following had a negative influence on fatalities: distance from home of 1–2 km (OR, 0.651) and ≥2 km (OR, 0.705); position of the road as intersection (OR, 0.622), first lane (OR, 0.667), and other (OR, 0.443); offending vehicle as motorcycle (OR, 0.120); collision part of the vehicle other than the wheel (OR, 0.508) (Table 2). The Hosmer–Lemeshow test indicated a good fit (*p* = 0.52), and the Nagelkerke R^2^ was 0.582.

### 3.3. Comparison of Severe and Mild Injury Cases

We investigated the non-fatal victims in terms of severe or mild injuries. The victim and collision characteristics were compared between the two groups (Table 3). Among victims with severe injuries, the prevalence of trunk injuries was significantly more common (*p* < 0.001); collisions mostly occurred at the weekend (*p* = 0.037), at night (*p* < 0.001), in rural areas (*p* < 0.016), and in the first lane of the road (*p* < 0.001). Regarding the offending vehicles in severely injured cases, there was a prevalence of passenger cars, light passenger cars, and trucks; the vehicles were mostly driving straight (*p* < 0.001). In terms of collision velocity, over 30 km/h accounted for 38.3% of victims with severe injuries; it accounted for 15.6% in those with mild injuries. Hit-and-run cases were significantly more frequent among victims with severe injuries than in those with mild injuries (*p* < 0.001). Regarding the colliding part of the vehicle, the wheel was significantly more common among victims with severe injuries than in those with mild injuries (*p* = 0.032).

To identify the variables that were independently associated with having severe injuries, we conducted multivariate logistic regression analysis using the forced input method. The following had a positive influence on having severe injuries: age (OR, 1.014); head or neck injuries (OR, 1.675) and trunk injuries (OR, 4.060) with extremities as the reference; night (OR, 1.470); collision velocity of 10–20 km/h (OR, 2.540), 20–30 km/h (OR, 3.700), 30–40 km/h (OR, 5.297); 40–50 km/h (OR, 5.719), and ≥50 km/h (OR, 5.244); and hit-and-run cases (OR, 2.628). The offending being a motorcycle (OR, 0.254) and the colliding part of the vehicle being other than the wheel (OR, 0.591) exerted a negative influenced on having severe injuries (Table 4). The Hosmer–Lemeshow test indicated a good fit (*p* = 0.387), and the Nagelkerke R^2^ was 0.300.

## 4. Discussion

The International Traffic Safety Data and Analysis Group aggregates international data on road crashes in 34 countries; according to the group, despite an overall positive trend, the rate of reduction in road deaths has slowed in recent years in most countries [2]. The average annual reduction in the number of road deaths was 3.3% in 1998–2008; however, it was only 2.3% in 2008–2018 [2]. Therefore, to improve traffic safety and address unresolved issues, there is a need for detailed investigations that contribute towards establishing effective preventive measures. In Japan, pedestrians lying on the road accounted for 8.3% of all pedestrian fatalities; however, few reports have dealt with such pedestrians [11,12]. The present study is the first to make a comprehensive analysis of the injuries of victims and vehicle characteristics with such collisions. To develop effective prevention strategies, the World Health Organization has stated that more comprehensive data about transport injuries are required [13]. The present study was conducted in Japan using the nationwide database of the NPA and is well in accordance with that recommendation.

This study confirmed that impact speed was a risk factor for fatalities and having severe injuries among pedestrians lying on the road: the ORs were extremely high. We found that the ORs increased with increasing impact velocity for fatalities and having severe injuries. Notably with a collision velocity of over 30 km/h, the ORs exceeded eight for fatalities and five for having severe injuries. Generally, speed management is a critical element in any road safety strategy: reducing speed is essential to achieve less frequent and less severe road crashes. Being run over has been considered critical regardless of the collision velocity; however, it is of great interest that decreasing collision velocity would be an effective way to reduce fatal or severe injuries for pedestrians lying on the road. Thus, there is a need for advances in vehicle technology for early recognition of pedestrians lying on the road and execution of immediate braking. For fewer road deaths and serious injuries, setting appropriate speed limits and enforcing them should be a core strategy. Lower speed limits are often in force in residential areas or around schools, typically 30 km/h [2]. Accordingly, strict speed limits are necessary in black spots to prevent collisions between vehicles and pedestrians lying on the road.

Regarding body regions, injuries to the head and neck or trunk were strong factors related to fatal or severe injuries. Both those regions include vital organs, and so we obtained extremely high ORs with extremities as the reference. If vehicles could avoid collisions with those body regions some fatalities and severe injuries might be prevented. We obtained information about the collision characteristics, but we were unable to ascertain why the offending vehicle could not avoid hitting the victims’ head and neck or trunk. Future in-depth investigations should focus on the interaction between pedestrians lying on the road and the offending vehicles.

Regarding the offending vehicles, motorcycles had a negative influence on fatalities and severe injuries; the ORs were 0.120 and 0.254, respectively. That is because motorcyclists are better able to avoid hitting objects on the road immediately after perceiving them. Among four-wheeled vehicles, trucks were a significant factor for fatalities: the OR was over two. Because of the heavy weight and wide wheels of trucks, victims would be subjected to considerable force if they were run over. According to the findings of forensic medicine, if a vehicle collides with a pedestrian lying on the road, in most cases the victim is struck by the vehicle floor rather than the wheels. Therefore, in this study, we focused on the minimum height of the vehicle floor. We found that a minimum height of the vehicle floor of under 18 cm was a significant factor for fatalities. However, we examined the height of the vehicle floor only for passenger cars, and we obtained data from less than half of them. Thus, the missing data may have somewhat influenced the results. According to both experimental and real-world crash studies of vehicle-pedestrian collisions where the pedestrians were upright, the risk of death or severe injuries generally depended on the type of vehicle [14,15,16]. Those studies found that light truck vehicles (including vans and utility vehicles) were associated with greater risk of pedestrian injury or fatality than passenger cars. We propose that vehicles with lower floors may be more harmful for pedestrians lying on the road. Further studies including the minimum height of the vehicle floor of all offending vehicles are required to confirm this issue.

We found that hit-and-run cases were a significant factor influencing both fatalities and severe injuries. Research has found that drivers are more likely to leave a collision scene in the following situations: where they recognized lower probability of being witnessed; when collisions occurred in the early morning; when there were no accompanying passengers; when they lacked a valid licence; when they were drunk; and with younger drivers [17,18,19]. Thus, preventing hit-and-run cases could decrease pedestrian fatality. Such measures as increasing the number of checkpoints for drunk driving and driving without a licence as well as the use of surveillance cameras should be considered.

Generally, infrastructure design and improvement are important traffic calming techniques. Helping prevent or restrict pedestrians from interacting with vehicles can eliminate conflicts [20]. Pedestrianization, which prevents pedestrians from accessing motorways and precludes vehicles from entering pedestrian zones, would appear to be an effective option [20]. That strategy could help reduce fatalities among pedestrians lying on the road: we found non-separated roads to be a major risk for such fatalities. Regarding the site of collision, we observed that a distance from the victim’s home of less than 1 km had a great influence on fatalities. Recently, ageing has become a nationwide problem in Japan, and older pedestrians with cognitive impairment are prone to wonder near their home and often suffer from vehicle collisions. It has been reported that a considerable number of pedestrians lying on the road were inebriated [5]. Those authors assumed that drunk pedestrians may have fallen asleep on their way home. Future research should examine the reasons for pedestrians lying on the road to help prevent such action. 

The present study has some limitations. First, it was based on information from the national database of the ITARDA. Although we obtained detailed information about the collisions, background information about the victims was limited. Thus, we did not analyze the reasons for lying on the road, such as drunkenness or loss of consciousness. More in-depth analyses for collecting detailed information about the victims are required. Second, we did not obtain detailed information regarding the interaction between the victims and the vehicles. The position of the victim’s body with respect to the traffic direction has a major effect on that person’s visibility. That information could help clarify why the offending vehicle was unable to avoid the collision. Further studies focusing on the interaction between the pedestrians lying on the road and the offending vehicles are required. Third, we did not obtain details of the collision location owing to personal information protection. However, if an investigation were made of places where collisions with pedestrians lying on the road have occurred multiple times, preventive measures with specially focused patrols could be implemented to help avoid such collisions. Further studies examining sites of such frequent occurrence are required.

## 5. Conclusions

According to the retrospective analysis with nationwide police database, we first investigated factors influencing fatalities and severe injuries of pedestrians lying on the road. Decreasing the collision velocity, avoiding collisions to the head and neck or trunk, and preventing hit-and-run cases would be effective in reducing fatal or severe injuries to pedestrians lying on the road. Previously, the height of obstacles on the road was identified as a factor that influenced the stopping sight distance [21,22]. According to one study, the height of objects affected the driver’s decision to stop immediately; a height of about 25 cm constitutes a physical hazard for drivers [21]. Thus, drivers unable to perceive pedestrians lying on the road are unable to stop safely.

The following steps should be taken to reduce the incidence of vehicle collisions with pedestrians lying on the road: adopting a multifaceted approach involving awareness of the risk of lying on the road by the media (including social media); appropriate education for pedestrians; developing a safety system to detect individuals lying on the road and avoiding collisions; improving the infrastructure of roads; and legislative changes.

## Figures and Tables

**Table 1 healthcare-09-01433-t001:** Victim and collision characteristics for fatal and non-fatal collisions.

	Total (*n* = 2452)	Fatal (*n* = 797)	Non-Fatal (*n* = 1655)	*p* Value
Victim characteristics				
Age	49.0 ± 20.7	55.3 ± 18.6	45.8 ± 20.9	<0.001
Sex (M/F)	2019/433	671/106	1348/307	0.096
Most severely injured body region				<0.001
Head and neck	767 (31.3%)	360 (45.2%)	407 (24.6%)	
Trunk	905 (36.9%)	355 (44.5%)	550 (33.2%)	
Extremities	706 (28.8%)	8 (1.0%)	698 (42.2%)	
All	74 (3.0%)	74 (9.3%)	0 (0%)	
Collision occurrence				
Day				0.533
Weekday	1529 (62.4%)	504 (63.2%)	1025 (61.9%)	
Weekend	923 (37.6%)	293 (36.8%)	630 (38.1%)	
Time				<0.001
Day	430 (17.5%)	31 (3.9%)	399 (24.1%)	
Night	2022 (82.5%)	766 (96.1%)	1256 (75.9%)	
Place				0.006
Urban	1311 (53.5%)	393 (49.3%)	918 (55.5%)	
Rural	546 (22.3%)	205 (25.7%)	341 (20.6%)	
Other	595 (24.3%)	199 (25.0%)	396 (23.9%)	
Distance from home				0.003
≤1 km	1348 (55.0%)	475 (59.6%)	873 (52.7%)	
1–2 km	266 (10.8%)	90 (11.3%)	176 (10.6%)	
>2 km	806 (32.9%)	225 (28.2%)	581 (35.1%)	
Unknown	32 (1.3%)	7 (0.9%)	25 (1.5%)	
Position on road				<0.001
Intersection	441 (18.0%)	124 (15.6%)	317 (19.2%)	
First lane	995 (40.6%)	428 (53.7%)	567 (34.3%)	
Second lane or above	128 (5.2%)	69 (8.7%)	59 (3.6%)	
Sngle lane road	530 (21.6%)	145 (18.2%)	385 (23.3%)	
Other	358 (14.6%)	31 (3.9%)	327 (19.8%)	
Collision characteristics				
Type of vehicle				<0.001
Passenger car	1397 (57.0%)	459 (57.6%)	938 (56.7%)	
Cargo	70 (2.9%)	20 (2.5%)	50 (3.0%)	
Light passenger car	501 (20.4%)	164 (20.6%)	337 (20.4%)	
Light cargo	113 (4.6%)	35 (4.4%)	78 (4.7%)	
Truck	147 (6.0%)	88 (11.0%)	59 (3.6%)	
Motorcycle	87 (3.5%)	6 (0.8%)	81 (4.9%)	
Other	137 (5.6%)	25 (3.1%)	112 (6.8%)	
Vehicle movement				<0.001
Going straight	1653 (67.4%)	681 (85.4%)	972 (58.7%)	
Starting	154 (6.3%)	14 (1.8%)	140 (8.5%)	
Turning left	256 (10.4%)	40 (5.0%)	216 (13.1%)	
Turning right	153 (6.2%)	28 (3.5%)	125 (7.6%)	
Reversing	101 (4.1%)	7 (0.9%)	94 (5.7%)	
Other	135 (5.5%)	27 (3.4%)	108 (6.5%)	
Collision velocity				<0.001
≤10 km/h	659 (26.9%)	62 (7.8%)	597 (36.1%)	
10–20 km/h	376 (15.3%)	71 (8.9%)	305 (18.4%)	
20–30 km/h	268 (10.9%)	69 (8.7%)	199 (12.0%)	
30–40 km/h	437 (17.8%)	202 (25.3%)	235 (14.2%)	
40–50 km/h	319 (13.0%)	194 (24.3%)	125 (7.6%)	
>50 km/h	249 (10.2%)	173 (21.7%)	76 (4.6%)	
Unknown	144 (5.9%)	26 (3.3%)	118 (7.1%)	
Hit-and-run				<0.001
Yes	431 (17.6%)	212 (29.6%)	219 (13.2%)	
No	2021 (82.4%)	585 (73.4%)	1436 (86.8%)	
Collision part of vehicle				0.623
Wheel	1210 (49.3%)	399 (50.1%)	811 (49.0%)	
Other	1242 (50.7%)	398 (49.9%)	844 (51.0%)	
Height of vehicle floor				0.077
≥18 cm	102 (4.2%)	26 (3.3%)	76 (4.6%)	
<18 cm	364 (14.8%)	133 (16.7%)	231 (14.0%)	
Unknown or inapplicable	1986 (81.0%)	638 (80.1%)	1348 (81.5%)	

**Table 2 healthcare-09-01433-t002:** Odds ratios and 95% confidence intervals for factors that affected fatal injuries.

	Odds Ratio	95% Confidence Interval
Victim characteristics		
Age	1.030	1.023–1.036
Sex		
M	Ref.	
F	0.827	0.597–1.114
Injured body region		
Extremities	Ref.	
Head or neck	90.221	42.844–189.99
Trunk	71.040	34.032–148.29
All	21,531	188.31–2,461,705
Collision occurrence		
Day		
Weekday	Ref.	
Weekend	1.320	1.038–1.679
Time		
Day	Ref.	
Night	2.003	1.218–3.294
Place		
Urban	Ref.	
Rural	0.822	0.605–1.117
Other	1.101	0.826–1.467
Distance from home		
<1 km	Ref.	
1–2 km	0.651	0.445–0.992
>2 km	0.705	0.536–0.926
Position on road		
Single lane road	Ref.	
Intersection	0.622	0.420–0.921
First lane	0.667	0.470–0.945
Second lane or above	0.653	0.368–1.158
Other	0.443	0.271–0.725
Collision characteristics		
Type of vehicle		
Passenger car	Ref.	
Cargo	0.871	0.389–1.949
Light passenger car	0.731	0.538–0.993
Light cargo	1.022	0.576–1.816
Truck	2.741	1.576–4.769
Motorcycle	0.120	0.048–0.303
Other	1.073	0.080–14.333
Vehicle movement		
Starting	Ref.	
Going straight	0.639	0.298–1.371
Turning left	0.924	0.423–2.020
Turning right	0.883	0.383–2.035
Reversing	0.591	0.200–1.752
Other	2.088	0.426–10.241
Collision velocity		
≤10 km/h	Ref.	
10–20 km/h	1.794	1.150–2.798
20–30 km/h	2.982	1.720–5.171
30–40 km/h	8.394	4.869–14.471
40–50 km/h	16.831	9.363–30.256
>50 km/h	18.639	9.970–34.846
Hit-and-run		
No	Ref.	
Yes	1.967	1.420–2.724
Collision part of vehicle		
Wheel	Ref.	
Other	0.508	0.396–0.651
Height of vehicle floor		
≥18 cm	Ref.	
<18 cm	1.984	1.028–3.826

**Table 3 healthcare-09-01433-t003:** Victim and collision characteristics for severe and mild injuries.

	Severe (*n* = 784)	Mild (*n* = 871)	*p* Value
Victim characteristics			
Age	48.6 ± 20.3	43.4 ± 21.2	0.320
Sex (M/F)	633/151	715/156	0.481
Most severely injured body region			<0.001
Head and neck	183 (23.3%)	224 (25.7%)	
Trunk	365 (46.6%)	185 (21.2%)	
Extremities	236 (30.1%)	462 (53.0%)	
Collision occurrence			
Day			0.037
Weekday	465 (59.3%)	560 (64.3%)	
Weekend	319 (40.7%)	311 (35.7%)	
Time			<0.001
Day	118 (15.1%)	281 (32.3%)	
Night	666 (84.9%)	590 (67.7%)	
Place			0.016
Urban	412 (52.6%)	506 (58.1%)	
Rural	184 (23.5%)	157 (18.0%)	
Other	188 (24.0%)	208 (23.9%)	
Distance from home			0.333
≤1 km	427 (54.5%)	446 (51.2%)	
1–2 km	84 (10.7%)	92 (10.6%)	
>2 km	259 (33.0%)	322 (37.0%)	
Unknown	14 (1.8%)	11 (1.3%)	
Position on road			<0.001
Intersection	144 (18.4%)	173 (19.9%)	
First lane	315 (40.1%)	253 (29.0%)	
Second lane or above	42 (5.4%)	17 (2.0%)	
Single lane road	169 (21.6%)	216 (24.8%)	
Other	115 (14.7%)	212 (24.3%)	
Collision characteristics			
Type of vehicle			<0.001
Passenger car	459 (58.5%)	479 (55.0%)	
Cargo	23 (2.9%)	27 (3.1%)	
Light passenger car	174 (22.2%)	163 (18.7%)	
Light cargo	39 (5.0%)	39 (4.5%)	
Truck	33 (4.2%)	26 (3.0%)	
Motorcycle	23 (2.9%)	58 (6.7%)	
Other	33 (4.2%)	79 (9.1%)	
Vehicle movement			<0.001
Going straight	533 (68.0%)	439 (50.4%)	
Starting	45 (5.7%)	95 (10.9%)	
Turning left	95 (12.1%)	121 (13.9%)	
Turning right	51 (6.5%)	74 (8.5%)	
Reversing	25 (3.2%)	69 (7.9%)	
Other	35 (4.5%)	73 (8.4%)	
Collision velocity			<0.001
≤10 km/h	183 (23.3%)	414 (47.5%)	
10–20 km/h	154 (19.6%)	151 (17.3%)	
20–30 km/h	111 (14.2%)	88 (10.1%)	
30–40 km/h	158 (20.2%)	77 (8.8%)	
40–50 km/h	87 (11.1%)	38 (4.4%)	
>50 km/h	55 (7.0%)	21 (2.4%)	
Unknown	36 (4.6%)	82 (9.4%)	
Hit-and-run			<0.001
Yes	135 (17.2%)	84 (9.6%)	
No	649 (82.8%)	787 (90.4%)	
Collision part of vehicle			0.032
Wheel	406 (51.8%)	405 (46.5%)	
Other	378 (48.2%)	466 (53.5%)	
Height of vehicle floor			0.221
≥18 cm	33 (4.2%)	43 (4.9%)	
<18 cm	121 (15.4%)	110 (12.6%)	
Unknown or inapplicable	630 (80.4%)	718 (82.4%)	

**Table 4 healthcare-09-01433-t004:** Odds ratios and 95% confidence intervals for factors that affected severe injuries.

	Odds Ratio	95% Confidence Interval
Victim characteristics		
Age	1.014	1.008–1.020
Sex		
M	Ref.	
F	1.236	0.918–1.665
Injured body region		
Extremities	Ref.	
Head or neck	1.675	1.243–2.257
Trunk	4.060	3.096–5.324
Collision occurrence		
Day		
Weekend	Ref.	
Weekday	0.966	0.764–1.222
Time		
Day	Ref.	
Night	1.470	1.096–1.972
Place		
Urban	Ref.	
Rural	1.193	0.881–1.615
Other	1.110	0.845–1.459
Distance from home		
<1 km	Ref.	
1–2 km	0.912	0.627–1.329
>2 km	0.941	0.734–1.206
Position on road		
Single lane road	Ref.	
Intersection	0.958	0.669–1.373
First lane	1.062	0.768–1.470
Second lane or above	1.438	0.714–2.898
Other	1.063	0.745–1.519
Collision characteristics		
Type of vehicle		
Passenger car	Ref.	
Cargo	1.028	0.523–2.023
Light passenger car	1.058	0.782–1.432
Cargo	1.136	0.654–1.973
Truck	1.253	0.673–2.332
Motorcycle	0.254	0.145–0.446
Other	0.332	0.049–2.235
Vehicle movement		
Starting	Ref.	
Going straight	0.828	0.512–1.340
Turning left	1.329	0.806–2.190
Turning right	0.933	0.520–1.676
Reversing	0.703	0.374–1.320
Other	1.292	0.415–4.025
Collision velocity		
≤10 km/h	Ref.	
10–20 km/h	2.540	1.804–3.575
20–30 km/h	3.700	2.370–5.774
30–40 km/h	5.297	3.333–8.419
40–50 km/h	5.719	3.310–9.882
>50 km/h	5.244	2.646–10.392
Hit-and-run		
No	Ref.	
Yes	2.628	1.725–4.004
Collision part of vehicle		
Wheel	Ref.	
Other	0.591	0.465–0.752
Height of vehicle floor		
≥18 cm	Ref.	
<18 cm	1.707	0.925–3.147

## Data Availability

The data presented in this study are available upon request from the corresponding author.

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
