# Peer review of "Factors Influencing Fatalities or Severe Injuries to Pedestrians Lying on the Road in Japan: Nationwide Police Database Study"

_healthcare, 2021, doi:10.3390/healthcare9111433_

Round 1

Reviewer 1 Report

This paper explores the factors associated to fatalities or severe injuries to pedestrians lying on the road in Japan. After careful reading, I shall raise my concern about two aspects of the manuscript:

1. There is a huge amount of previous literature which assess factors associated to fatalities or severe injuries in pedestrians. What I really miss here is some context to help the potential readers to understand why the authors have decided to study specifically those pedestrians lying on the road. Is this a common phenomenon in Japan? If so, why does it happen? (people willing to suicide, “challenges” among teenagers…?). This information is not registered in police databases of other countries, so readers from there may find this topic “weird” and unnecessary. Hence, I believe that clarifying this point would help to justify why this type of accidents deserve attention among other pedestrian crashes. Also, it would increase the interest of the potential audience for this manuscript.

2. Analysis: authors have performed two multivariate logistic regression analyses. One is for identifying factors associated to death, and the other one is for identifying factors associated to severe injuries. Instead, have they considered performing a multinomial logistic regression? (the outcome variable can be considered to have three categories: death, severe injuries and mild injuries). I think this alternative could be more elegant and informative.

More minor comments:

1. Definition of death: for some years now, it has been preferred to consider it within 30 days of the collision (as WHO does). Why have the authors chosen 24h instead?

2. Police databases, like the one the authors have used, sometimes have a considerable amount of missing data in some variables. Here, it happens specially in the height of the vehicle floor (more than 80% of missing data). How could this affect the results? Maybe it would be adequate using multiple imputation.

3. The p-value column is unnecessary in tables 2 and 4, as 95% CI are shown.

4. Page 10, Discussion: “we observed that hit-and-run cases were a significant factor for causing both fatality and severe injuries”. Please, be careful when talking about causality (specially in a cross-sectional study like this). Obviously, the fact that a driver run away after colliding with a pedestrian does not cause the pedestrian to die (the collision does).

Author Response

Thank you for the thoughtful and constructive feedback you provided regarding our manuscript. In accordance with your suggestion, we have revised the manuscript. We are grateful for the time and energy you expended on our behalf.

This paper explores the factors associated to fatalities or severe injuries to pedestrians lying on the road in Japan. After careful reading, I shall raise my concern about two aspects of the manuscript:

  1. There is a huge amount of previous literature which assess factors associated to fatalities or severe injuries in pedestrians. What I really miss here is some context to help the potential readers to understand why the authors have decided to study specifically those pedestrians lying on the road. Is this a common phenomenon in Japan? If so, why does it happen? (people willing to suicide, “challenges” among teenagers…?). This information is not registered in police databases of other countries, so readers from there may find this topic “weird” and unnecessary. Hence, I believe that clarifying this point would help to justify why this type of accidents deserve attention among other pedestrian crashes. Also, it would increase the interest of the potential audience for this manuscript.

   Some studies of pedestrian injuries have underlined the importance of investigating the pre-crash behavior of the pedestrians involved: we have included those as references in the revised manuscript. Such collisions have been particularly examined by forensic practitioners; however, until 2012, no comprehensive statistics in that regard existed for Japan. Thus, many practitioners have stressed the importance of collecting data about collisions involving pedestrians lying on the road in that country. To promote in-depth investigation of vehicle-pedestrian collisions, Japan’s National Police Agency has since 2012 included the category of collisions involving pedestrians lying on the road. Recently, a detailed investigation found that pedestrians lying on the road accounted for 8.3% of all pedestrian fatalities for 2012–2016; thus, the numbers are considerable. Accordingly, the present study was undertaken toward reducing fatal pedestrian injuries.

              We have added a description to explain this situation in the Introduction (lines 56–76). We addressed the importance of this issue in the first paragraph of the Discussion in the original version of our paper. Regarding the reasons for pedestrians lying on the road, we were unable to analyze that owing to lack of information, as indicated among the limitations discussed in the original version. However, we have now included possible causes as a result of one report presented in the Introduction (lines 65–67).

  1. Analysis: authors have performed two multivariate logistic regression analyses. One is for identifying factors associated to death, and the other one is for identifying factors associated to severe injuries. Instead, have they considered performing a multinomial logistic regression? (the outcome variable can be considered to have three categories: death, severe injuries and mild injuries). I think this alternative could be more elegant and informative.

  Thank you for this very useful comment. As you note, multinomial logistic regression analysis is applied for more than two nominal dependent variables. In the present study, we considered that the categories of death, severe injury, and mild injury were not equivalent. In line with most previous studies, we categorized all the injuries as fatal or non-fatal to identify the significant factors relating to death. However, in our subsequent analysis, we investigated severely or mildly injured pedestrians: all the patients had been transported to medical facilities and received care as appropriate. The fatal cases included pedestrians who suffered instant death and were not transported to medical facilities or did not receive medical care. Thus, we performed two multivariate logistic regression analyses. However, in light of your very good suggestion, we would like to compare such results in the future.

More minor comments:

  1. Definition of death: for some years now, it has been preferred to consider it within 30 days of the collision (as WHO does). Why have the authors chosen 24h instead?

We fully understand your concern. The Japanese government generally defines collision death as a fatality occurring within 24 hours of a collision. In line with international trends, fatalities occurring within 30 days of a collision have also recently been recorded in Japan. However, statistics related to fatalities occurring within 30 days are limited to major items and are used for international compassions in the IRTAD database. As a result, we were unfortunately unable to obtain the numbers of fatalities within 30 days of collisions involving pedestrians lying on the road. However, this is an area of concern: we would like to propose that the government obtain all data relating to fatalities occurring within 30 days of collision.

  1. Police databases, like the one the authors have used, sometimes have a considerable amount of missing data in some variables. Here, it happens specially in the height of the vehicle floor (more than 80% of missing data). How could this affect the results? Maybe it would be adequate using multiple imputation.

We found no missing data except those related to the height of the vehicle floor. As the reviewer notes, we obtained those data for some of the cases. Those data applied only to passenger cars; further, with such cars, we could obtain the data from less than half of the cases. As the reviewer observes, the missing data may have somewhat affected the results. In line with the reviewer’s comments, we have rewritten part of the Materials and Methods (lines 120–122) and Discussion (lines 254–256, 261–263) sections regarding the height of vehicle floor and potential harm. We apologize for not having selected the suggested method of multiple imputation for our analyses owing to our lack of knowledge. Following your suggestion, we will study that method and apply it in further analyses. We are very grateful for this valuable suggestion.

  1. The p-value column is unnecessary in tables 2 and 4, as 95% CI are shown.

Following the reviewer’s suggestion, we have omitted the column showing 95% CIs in Tables 2 and 4.

  1. Page 10, Discussion: “we observed that hit-and-run cases were a significant factor for causing both fatality and severe injuries”. Please, be careful when talking about causality (specially in a cross-sectional study like this). Obviously, the fact that a driver run away after colliding with a pedestrian does not cause the pedestrian to die (the collision does).

In line with your suggestion, we have rewritten the text in question (lines 264–265).

Reviewer 2 Report

Interesting article on one of the problems related to road safety.

Please show an example record from the ITARDA database to show the scope of collected data. 

Why is the height of the vehicle floor described in table 1 as "unknown or not applicable" and in table 2 as "other"? Please standardize. 

In conclusions, please refer to the issue of stopping sight distance (including the height of the obstacle used in such calculations) in various road conditions.

Author Response

Thank you for the thoughtful and constructive feedback you provided regarding our manuscript. In the following sections, you will find our responses to each of your points and suggestions. We are grateful for the time and energy you expended on our behalf.

Interesting article on one of the problems related to road safety.

Please show an example record from the ITARDA database to show the scope of collected data.

  Following your suggestion, we tried to obtain an example of a record from the database. However, we were unable to receive such a record related to a victim owing to legal considerations in Japan. We are sorry that we are unable to comply with your suggestion.

Why is the height of the vehicle floor described in table 1 as "unknown or not applicable" and in table 2 as "other"? Please standardize.

  In accordance with your suggestion, we have corrected the description in Table 3.

In conclusions, please refer to the issue of stopping sight distance (including the height of the obstacle used in such calculations) in various road conditions.

  In line with your suggestion, we have added a description about the stopping sight distance and height of obstacles in the Discussion, and we have provided two additional references (lines 305–309).

Reviewer 3 Report

This manuscript addresses the influencing factors of reducing fatalities or injuries of pedestrians lying on the road using Japan's Nationwide Police Database. In principle, the project is an interesting and important study. I have several concerns regarding the manuscript.

* Please add the Literature Review part and add some important references.

* Why do the authors define death as the fatality occurring within 24 hours of the collision? 

* Please check and improve Table 1, Table 2, Table 3 and Table 4.

* There are significant language shortcomings.

Author Response

Thank you for the thoughtful and constructive feedback you provided regarding our manuscript. In the following sections, you will find our responses to each of your points and suggestions. We are grateful for the time and energy you expended on our behalf.

This manuscript addresses the influencing factors of reducing fatalities or injuries of pedestrians lying on the road using Japan's Nationwide Police Database. In principle, the project is an interesting and important study. I have several concerns regarding the manuscript.

* Please add the Literature Review part and add some important references.

  Following your suggestion, we have added content related to a literature review in the Introduction, and we have provided six references (lines 42–63).

* Why do the authors define death as the fatality occurring within 24 hours of the collision?

  We fully understand your concern. The Japanese government generally defines death as a fatality occurring within 24 hours of a collision. In line with international trends, fatalities occurring within 30 days of a collision have also recently been recorded in Japan. However, statistics related to fatalities occurring within 30 days are limited to major items and are used for international compassion in the IRTAD database. As a result, we were unfortunately unable to obtain the numbers of fatalities within 30 days of collisions involving pedestrians lying on the road. However, this is an area of concern: we would like to propose that the government obtain all data relating to fatalities occurring within 30 days of collision.

* Please check and improve Table 1, Table 2, Table 3 and Table 4.

  Following your suggestion, we have checked and improved the tables, and we have had them edited by a professional editing company using native English speakers.

* There are significant language shortcomings.

  Before submitting the original version, our paper did in fact undergo professional text editing by native English speakers. In line with your comment, the revised version of the paper similarly underwent professional text editing in the same manner.

Round 2

Reviewer 3 Report

I have presented my comments and suggestions in the first revision. I accept the authors' answers.